# Survival Rates of Endodontically Treated Posterior Teeth Restored with All-Ceramic Partial-Coverage Crowns: When Systematic Review Fails

**DOI:** 10.3390/ijerph19041971

**Published:** 2022-02-10

**Authors:** Marco Ferrari, Edoardo Ferrari Cagidiaco, Denise Irene Karin Pontoriero, Carlo Ercoli, Kostantinos Chochlidakis

**Affiliations:** 1Department of Prosthodontics and Dental Materials, University of Siena, 53100 Siena, Italy; edoardo.ferrari.cagidiaco@gmail.com (E.F.C.); denise.pontoriero@unisi.it (D.I.K.P.); 2Department of Prosthodontics, Eastman Institute for Oral Health, University of Rochester, Rochester, NY 14627, USA; Carlo_Ercoli@URMC.Rochester.edu (C.E.); kchochlidakis@URMC.Rochester.edu (K.C.)

**Keywords:** partial coverage crowns, all-ceramic restorations, endodontically treated teeth, fiber posts, posterior teeth

## Abstract

Background: To determine the survival rates of endodontically treated posterior teeth (EDPT) restored with partial coverage all-ceramic crowns with or without the use of fiber posts. Methods: MEDLINE and Cochrane searches were conducted in order to identify Randomized Clinical Trials (RCTs) related to endodontically treated posterior teeth restored with partial coverage crowns. The search period was extended until February 2020 and only in vivo, human, and studies in the English language were included. A manual search was also conducted and additional articles, if found, were included in the database. Results: The initial search for the selected databases identified 495 studies, which were all screened for inclusion through titles, abstracts and full-text reading. Out of these 495 studies, only one article met the eligibility criteria and was included in this systematic review. Statistical analysis could not be performed. Conclusions: Only one RCT was identified in this systematic review. More clinical evidence is necessary to assess the survival rate of EDPT with partial-coverage crowns. This systematic review failed because it did not find scientific evidence to support the use of indirect bonded restorations on EDPT.

## 1. Introduction

The best clinical procedure to restore endodontically treated teeth (ETT) is still under discussion. The use of indirect bonded restorations transformed clinical behavior, saving tooth structure and becoming widely accepted by practitioners. Evidence is needed by RCTs to confirm if bonded restorations can replace cemented restorations clinically. In order to obtain scientific evidence, a systematic review should be performed based on at least three RCTs.

During the fabrication of the definitive restoration on ETT, post/dowel and core systems are frequently required, especially when the remaining tooth structure is inadequate to provide retention and resistance for the definitive restoration [1]. Indeed, tooth-structure loss due to caries, trauma, or previous restorations can decrease the available tooth structure and without the use of a post and a core build-up, may not provide adequate retention of the final prosthesis [2,3]. In these clinical situations, intra-radicular posts are often recommended [1,4]. The selection of the most suitable post/dowel and core system can be challenging, and several different techniques and materials have been used in clinical practice [1].

Cast posts and cores have been widely used throughout the years and have long been considered the gold standard [2,3]. However, a high prevalence of complications has been reported in the literature such as a loss of retention and of root fractures [5,6]. Fiber posts have therefore been suggested as a more conservative alternative to cast posts due to the similarity between their modulus of elasticity and that of dentin [7,8,9].

The loss of pulp vitality and endodontic treatment also cause biomechanical changes to the ETT [10]. While several factors affect the clinical performance of ETT restored with posts, there seems to be a strong correlation with the preservation of tooth structure (the number of the remaining walls and/or the amount of coronal residual structure), and, therefore, an adequate ferrule [6,7,10,11,12]. In addition, the type of post, the position of the tooth in the arch, the number of interproximal contacts and the type of the definitive prosthesis could also influence the prognosis of the ETT and their supporting teeth [13]. Specifically, fiber posts have an elastic modulus (25–57 GPa) closer to that of dentine (18 GPa) and distribute stresses to the surrounding tooth structure [6,8,9,14,15,16,17]. ETT with interproximal contacts appear to have a higher rate of survival, likely due to stress distribution and support from the neighboring teeth [13,18,19]. The effect of tooth type and position is rather more controversial [20]; indeed, while anterior teeth and premolars could be potentially more prone to non-axial loading compared to molars, in a clinical study, no difference was identified between anterior and posterior teeth [21]. In contrast, two studies reported that anterior teeth have a higher failure risk [13,22], while one prospective clinical study [23] and two in vitro studies [24,25] reported that premolars failed more frequently than anterior teeth [23,24,25]. Similarly, when the type of definitive restoration was considered, no correlation between failure rates of fiber post-restored teeth and the type of prosthesis (single crown or fixed dental prosthesis) could be established [13,26].

From the data available, it is clear that the most important factor underlying the prognosis of ETT is the preservation of tooth structure. In light of this, an adhesive, partial-coverage restoration, either ceramic or composite, Refs. [27,28,29,30,31] has the advantage of requiring less tooth structure removal than a complete coverage crown/retainer. Conversely, a more conservative method of tooth preparation can also preserve available enamel, which significantly enhances the predictability of bonded restorations [32,33]. While the fabrication of partial coverage restorations has been the subject of several case reports and series, evidence-based information is lacking as to the need for post/dowel and core reconstructions in association with these definitive prostheses.

Therefore, this systematic review aimed to evaluate the available evidence related to the survival rates of ETPT restored with partial-coverage ceramic crowns with or without the use of fiber posts.

## 2. Materials and Methods

This systematic review was performed according to the Preferred Reporting Items for Systematic Reviews and Meta-Analysis (PRISMA). The following PICO was designed:

P: Endodontically treated teeth receiving partial-coverage ceramic crowns

I: Treated with posts

C: Not treated with posts

O: Survival of teeth and partial-coverage ceramic crowns

A MEDLINE, Scopus, Google Scholar and a Cochrane search were conducted to identify randomized controlled clinical trials (RCTs) related to endodontically treated posterior teeth restored with partial-coverage crowns. The search strategy was the same for both MEDLINE and Cochrane libraries and include the following keyword combinations: “fiber post” OR “fiber posts” OR “fiber-reinforced post” OR “fiber-reinforced posts” AND “root filled teeth” OR “root-filled tooth” OR “endodontically treated teeth” OR “endodontically treated tooth” OR “fiber post restored teeth” AND “adhesive partial crowns” AND “ indirect restoration”.

Search filters were the publication date (from January 1990 up to July 2021), language (English), species (human), and article type (clinical trial). Two independent reviewers screened all titles and abstracts to determine whether the following inclusion criteria were met:In vivo.Adult human subjects.Direct quantitative assessment (success, survival, failure and/or complications) of the role of ceramic partial-coverage crowns.Prospective, randomized.Endodontically treated teeth restored with or without fiber post/dowel and ceramic partial-coverage crowns.

Whenever it was not possible to make this determination from the title and abstract, the full-text article was examined. Subsequently, all relevant articles were obtained, and two reviewers decided if they met the inclusion criteria. A manual search of the relevant references was also performed to identify other potentially relevant articles.

## 3. Results

The electronic database searches identified 495 articles. After the removal of irrelevant and duplicate articles, 115 studies remained. From the remaining relevant articles, the examination of titles and abstracts revealed that 67 were in vitro studies, four were finite element analyses, three were case reports, and nine were literature reviews. Five studies were also retrieved from the references of the selected articles. Finally, 27 clinical studies were read in full text for eligibility criteria assessment and 26 of them did not meet the inclusion criteria so did not qualify. These 26 excluded studies mainly concentrated on the type of core build-up of ETT, and those teeth were restored with cemented full-coverage crowns [11,12,13,15,17,19,20,21,22,23,34,35,36,37,38,39,40,41,42,43,44,45,46,47,48,49,50] (Figure 1). Only one randomized clinical study met all the required criteria and was included in the review as a relevant study [31]. This paper was a randomized controlled prospective clinical trial with a 3-year follow up about posterior partial crowns out of lithium disilicate (LS2) with or without posts. The paper showed only two failures for the group of premolars restored without posts, but no significant differences were found between teeth in which posts were luted and teeth restored without posts.

## 4. Discussion

The restoration of endodontically treated teeth with increased structure loss might be a challenge. In many situations the use of a post and core is required to support the core, which further retains the restoration. Depending on the level of tooth-structure loss, the remaining number of walls, ferrule effect, type of restoration, and the root anatomy, a post might be necessary [1,51]. After the endodontic treatment, the ETT is restored with either direct restorations, full-coverage crowns, fixed partial dentures or partial-coverage crowns [11,31,52,53]. The decision for the definitive restoration depends on the remaining amount of tooth structure after the endodontic therapy, finances, longevity as well as mechanical properties of the restoration. This clinical procedure is very new in our specialty, and very important from the medical perspective and for developing protocols in future.

Direct restorations after the endodontic treatment may be mainly used as provisional (intermediate step) restoration before the definitive restoration [27]. In some cases, direct restorations are used as definitive restorations on ETT. Even though the chair time and associated costs are less than a direct restoration, the mechanical behavior of the direct restorations could lead to possible fractures and/or chipping of the coronal part within years under clinical service, and especially in posterior teeth [52]. When the remaining coronal tooth structure is adequate, a bonded direct restoration could be possibly indicated. A previous study showed that more than 70% of posterior ETT have already lost more than 2/3 of their coronal tooth structure and a bonded restoration would not be an option [3].

The use of full-coverage crowns to restore single posterior ETT, either with or without a post, has been widely documented [10,11]. Most of the available studies report on full-coverage restorations cemented with traditional cements, such as zinc phosphate or glass-ionomer based cements, and most of the crowns already have a retentive and resistance form [10,11,12,13,14,15,16,17]. However, the cement remains the weakest part of this system and dislodgement and/or decementation of the crown is one of the main reasons for failure [10,11]. Recent studies reported on the use of self-adhesive cements to bond all-ceramic crowns on posterior ETT, but the mechanical behavior of this system is still unclear [51]. In most of the cases of endodontically treated premolars, the use of a post and core might be indicated to retain the definitive restoration. Conversely, endodontically treated molars may be restored with or without a post, depending on the remaining coronal tooth structure and ferrule effect [54,55]. Although the use of onlays, overlays and partial-coverage crowns in restoring posterior ETT is well accepted by clinicians, the available evidence is scarce.

This systematic review aimed to summarize clinical evidence regarding the restoration of posterior ETT with partial-coverage all-ceramic crowns with or without the use of fiber posts. Only one recently published study met the eligibility criteria of this systematic review [31]. This study assessed the influence of posts in posterior ETT with lithium disilicate partial-coverage crowns. After 3 years of clinical service, the placement of fiber posts did not seem to be significant for their success and survival rate. Although no statistically significant differences were found between molars and premolars, the results showed that premolars had a greater risk of failure. Two maxillary premolars failed, and in both cases, group function occlusion was observed. This observation suggests the use of posts on the upper premolars when included in a group of lateral functions, and to observe more details on the occlusion of each patient also when a single endodontically treated posterior tooth has to be restored [31].

Because only one RCT was found and evaluated, it was not possible to perform a conventional systematic review; this aspect opens the discussion because many practitioners are already using indirect bonded restorations on endodontically treated teeth, with or without posts. Therefore, whether a well-accepted procedure not yet scientifically proved by RCTs can be ethically and clinically acceptable for practitioners and patients should be discussed. Of course, performing a clinical trial is a lot of work and it will only be possible to check available data and perform a systematic review by increasing the number of RCTs.

A recent systematic review on the role of occlusion with adhesively cemented partial coverage crowns concluded that no randomized clinical trials were available [56]. Another systematic review investigated the role of occlusion on teeth restored with fiber posts and reported a lack of evidence on this topic [57]. The type of occlusion could be a contributing factor in the prognosis of endodontically treated teeth but there is no good evidence investigating the role of occlusion in ETT. In some clinical scenarios, it should be pointed out that adhesively cemented partial-coverage crowns could be the final definitive restorations of a single posterior in single ETT, of several multiple ETT in a quadrant or part of a more complex rehabilitation till a or a full-mouth rehabilitation reconstruction [57,58,59,60,61,62,63,64].

The use of adhesive partial crowns can improve the reliability and predictability of the prosthodontics treatment, especially in the treatment planning of young not worn posterior teeth, because of its conservative approach, saving the coronal structure and keeping margins on the enamel. The included sample teeth used by Ferrari et al. [38] were all in similar clinical situations in which occlusal wear was very light, not present or absent [38]. In most prosthetic therapies (relatively small amounts of restorative treatments, e.g., up to two or three units of crown or bridge work), the static position of the occlusion between the arches and the dynamic occlusal relationship should not be altered during treatment [65,66,67]. No information on endodontically treated posterior teeth with severe wear is available.

However, the use of adhesive partial-coverage crowns on endodontically treated posterior teeth has several advantages such as a more conservative preparation, the presence of a wide part of the margins in the enamel and a different behavior under occlusal loading. The type of tooth preparation and the shape of the cavity are determined by the presence of old restorations, the amount of caries, the extent of the endodontic cavity, and type of build-up used [27,28,29,30]. Regarding the mechanical behavior of bonded partial-coverage crowns, the bonded procedure and the adhesive materials used to lute them create a sort of monoblock with the abutment and help to dissipate the occlusal loading along with the external structure of the crown and the roots.

Ferrari et al. used lithium disilicate partial crowns to restore ETPT but several other esthetic materials such as reinforced resin composite and porcelain are available [31]. There is also an urgent need for RCTs with a longer observation time than 3 years and testing adhesive partial crowns made with different prosthodontic materials in patients with different types of occlusion. However, further RCTs, a longer observation time and possibly aesthetic partial crowns fabricated with different materials are needed to understand how different anterior guides and occlusal determinants could influence the prognosis and the therapeutic choice of ETT [26,38].

In this systematic review, only one study related to the survival rates of endodontically treated posterior teeth restored with all-ceramic partial-coverage crowns was found. One clinical study is not sufficient to make any definitive conclusion, however. Further RCTs are needed to clarify the role of some essential factors regarding the inclusion of a post, the design of all-ceramic partial crowns and the type of ceramic materials, to provide relevant clinical indications for the treatment planning for ETT posterior reconstructions.

## 5. Conclusions

This article shows that an already well-accepted clinical procedure based on in vitro evaluation data might not be supported by scientific evidence yet.

## Figures and Tables

**Figure 1 ijerph-19-01971-f001:**
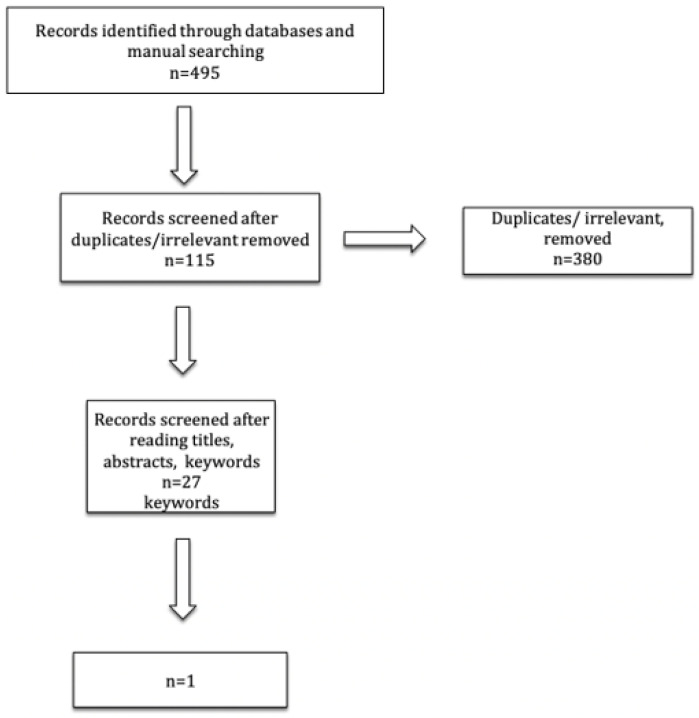
Workflow of the paper screening process.

## Data Availability

The data presented in this study are available on request from the corresponding author. The data are not publicly available due to privacy and protection of intellectual property.

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
