# Peer review of "Survival Rates of Endodontically Treated Posterior Teeth Restored with All-Ceramic Partial-Coverage Crowns: When Systematic Review Fails"

_ijerph, 2022, doi:10.3390/ijerph19041971_

Round 1

Reviewer 1 Report

Thanks for authors' efforts of reviewing the published work and indicating the potential lacking of clinical evidence problem of a current dental restoration technique which is important. Here are some concerns from me about this manuscript:

  1. Introduction - Please give the full name of ETT. According to the "ETPT" in the abstract, I suppose the ETT representing "endodontically treated teeth". Is it correct? Please either make the phase and abbreviation consistent or give the full name of ETT in the Introduction part.
  2. Introduction- What is the "beer distribute stress" and "beer survival"? It looks like that these are not common concepts. But maybe I am wrong. Could authors add concepts in the text?
  3. Introduction - The last sentence of Introduction, i.e."Therefore, the aim of ... the use of fiber posts.", is almost exactly same as one sentence in the Discussion part, i.e. "The aim of this...the use of fiber posts."(the 1st sentence of the 4th paragraph). Please consider modify this.
  4. Conclusion - the "This article shows that an already well-accepted clinical procedure might be not supported by scientific evidence yet." might not be quite appropriate as it was mentioned that there are in vitro scientific research about this technique. Please revise this conclusion and make it more accurate.
  5. Languages need to be revised.

Author Response

Reviewer 1.

Thanks for authors' efforts of reviewing the published work and indicating the potential lacking of clinical evidence problem of a current dental restoration technique which is important. Here are some concerns from me about this manuscript:

  1. Introduction - Please give the full name of ETT. According to the "ETPT" in the abstract, I suppose the ETT representing "endodontically treated teeth". Is it correct? Please either make the phase and abbreviation consistent or give the full name of ETT in the Introduction part.

The article was edited as suggested (see along the introduction section)

  1. Introduction- What is the "beer distribute stress" and "beer survival"? It looks like that these are not common concepts. But maybe I am wrong. Could authors add concepts in the text?

The reviewer was right to mention that the ‘beaer’ concept was not so popular and might b misunderstood by the reader. For that we decided to cut the word ‘beaer’ from the text.

  1. Introduction - The last sentence of Introduction, i.e."Therefore, the aim of ... the use of fiber posts.", is almost exactly same as one sentence in the Discussion part, i.e. "The aim of this...the use of fiber posts."(the 1st sentence of the 4th paragraph). Please consider modify this.

The sentence in the Discussion section was modified accordingly.

  1. Conclusion - the "This article shows that an already well-accepted clinical procedure might be not supported by scientific evidence yet." might not be quite appropriate as it was mentioned that there are in vitro scientific research about this technique. Please revise this conclusion and make it more accurate.

The conclusion was modified accordingly.

  1. Languages need to be revised.

The text was read by a native language dentist (End of first paragraph of Discussion).

Reviewer 2 Report

The results of this cohort systematic review are shocking, I agree. And the overall text of the article can be considered a little short, but actually regarding this type of study, you can not write a lot of words. In this cohort systematic review, after cutting the variables, only one study remained to be analysed. And the conclusions are right, underlined also in the title of the article.   Points of strength would be: - topic - very new in our specialty, very important from medical perspective and future developing protocols - number of studies analysed - 495 - a LOT of work, and this, unfortunately, can be checked only if someone will redo all the work.

Author Response

Reviewer 2.                                                     

The results of this cohort systematic review are shocking, I agree. And the overall text of the article can be considered a little short, but actually regarding this type of study, you can not write a lot of words. In this cohort systematic review, after cutting the variables, only one study remained to be analysed. And the conclusions are right, underlined also in the title of the article.  Points of strength would be: - topic - very new in our specialty, very important from medical perspective and future developing protocols - number of studies analysed - 495 - a LOT of work, and this, unfortunately, can be checked only if someone will redo all the work.

In the Discussion section we reported some considerations made by the reviewer.

Reviewer 3 Report

The aim of this systematic review was to evaluate the available evidence related to the survival rates of endodontically treated posterior teeth restored with partial-coverage ceramic crowns with or without the use of fiber posts.

The very small number of included studies (one included study) with short follow-up was further limited to draw definite conclusions although the topics mentioned about systematic review fails.

The systematic review should follow the PRISMA guidelines, and only one number of included study cannot give any valid conclusion. Otherwise, it should not be prepared as a systematic review paper. But I found that there are some important parts missing in the draft as follows. 

1) Focused question? 

2) Data extraction? 

3) Risk of bias assessment? 

Author Response

Reviewer 3.

The aim of this systematic review was to evaluate the available evidence related to the survival rates of endodontically treated posterior teeth restored with partial-coverage ceramic crowns with or without the use of fiber posts.

The very small number of included studies (one included study) with short follow-up was further limited to draw definite conclusions although the topics mentioned about systematic review fails.

The systematic review should follow the PRISMA guidelines, and only one number of included study cannot give any valid conclusion. Otherwise, it should not be prepared as a systematic review paper. But I found that there are some important parts missing in the draft as follows. 

  • Focused question? 

Focused question was reported at the beginning of Materials and Methods.

  • Data extraction? 

Data extraction was reported along Materials and Methods

  • Risk of bias assessment? 

Accordingly with the fact that only one paper was considered valid to be included in the systematic review, no risk of bias at all.

Reviewer 4 Report

The present research aimed to investigate the survival rates of endodontically treated posterior teeth restored (ETPT) with all-ceramic partial crowns with or without fiber-posts. The research question is of high clinical interest since, in daily practice, dentists restore ETPT with various types of restorations, although there is insufficient data to support one method or another. Therefore, this type of review papers are necessary for providing clinicians with evidence-based findings to support their choice.

However, some minor concerns should be cleared before the paper can be considered for publication:

  1. The keywords are not representative of the present manuscript. Consider including the most frequent words in the manuscript like ‘partial coverage crowns’, ‘all-ceramic restorations’, ‘endodontically treated teeth’, ‘fiber posts’, ‘posterior teeth’, etc.
  2. The search period must include the beginning and the ending of the time frame, not only until the research was conducted. Also, these intervals must coincide in the abstract and the manuscript.
  3. Check spelling and grammar. See, for example, the usage of the term “beer survival” in the introduction section.

Author Response

Reviewer 4.

The present research aimed to investigate the survival rates of endodontically treated posterior teeth restored (ETPT) with all-ceramic partial crowns with or without fiber-posts. The research question is of high clinical interest since, in daily practice, dentists restore ETPT with various types of restorations, although there is insufficient data to support one method or another. Therefore, this type of review papers are necessary for providing clinicians with evidence-based findings to support their choice.

However, some minor concerns should be cleared before the paper can be considered for publication:

  1. The keywords are not representative of the present manuscript. Consider including the most frequent words in the manuscript like ‘partial coverage crowns’, ‘all-ceramic restorations’, ‘endodontically treated teeth’, ‘fiber posts’, ‘posterior teeth’, etc.

Keywords were changed accordingly

  1. The search period must include the beginning and the ending of the time frame, not only until the research was conducted. Also, these intervals must coincide in the abstract and the manuscript.

The search period was reported in the Materials and Methods section.

  1. Check spelling and grammar. See, for example, the usage of the term “beer survival” in the introduction section.

Done

Round 2

Reviewer 1 Report

OK with publishing.

Author Response

The article was review by a native language teacher and some minor spells and the style were modified accordingly.

The Conclusion section was changed as the Editor suggested and part of the previous conclusion section was moved up as last sentence of Discussion section, as suggested.
